# Social Acceptance of Mobile Health among Young Adults in Japan: An Extension of the UTAUT Model

**DOI:** 10.3390/ijerph192215156

**Published:** 2022-11-17

**Authors:** Jianfei Cao, Karin Kurata, Yeongjoo Lim, Shintaro Sengoku, Kota Kodama

**Affiliations:** 1Graduate School of Technology Management, Ritsumeikan University, Ibaraki 567-8570, Japan; 2Graduate School of Corporate Business, Ritsumeikan University, Ibaraki 567-8570, Japan; 3School of Environment and Society, Tokyo Institute of Technology, Tokyo 152-8550, Japan; 4Center for Research and Education on Drug Discovery, The Graduate School of Pharmaceutical Sciences, Hokkaido University, Sapporo 060-0812, Japan

**Keywords:** mHealth, health consciousness, UTAUT model, Japanese young adults, mobile health

## Abstract

The unprecedented development of information and communication technologies has opened up immense possibilities in the field of health care. Mobile health (mHealth) is gaining increasing attention as an important technology for solving health-related problems. Although a high rate of smartphone usage among young people in Japan has been identified, smartphone usage for health management is not high. As Japanese youth are important potential users of mHealth, it is necessary to explore theories that influence the behavioral intention of Japanese youth to adopt mHealth. This study conducted a questionnaire survey in a Japanese university and collected 233 valuable responses. This study was adapted and extended from the unified theory of acceptance and use of technology (UTAUT) model to measure eight constructs: health consciousness, social influence, facilitation conditions, perceived risk, trust, performance expectancy, effort expectancy, and behavioral intention. Structural equation modeling was used for hypothesis testing. We found that trust, performance expectancy, and effort expectancy directly influenced the behavioral intention to use mHealth. Health consciousness and social influence indirectly influence behavioral intention through trust and performance expectancy. Facilitation conditions indirectly influenced behavioral intention through effort expectancy. This study makes a vital theoretical contribution to policymakers and product developers for the further diffusion of mHealth among young people in Japan.

## 1. Introduction

The unprecedented popularity of smartphones has made people’s lives highly convenient. The Japanese Ministry of Internal Affairs and Communications White Book (2021) reported a smartphone ownership rate of 86.8% in Japan in 2020 [1]. Moreover, with the rapid development of information and communication technology (ICT), healthcare professionals worldwide are actively exploring the possibility of using smartphones for health management [2,3,4]. In Japan, the number of diseases (e.g., hypertension, diabetes, etc.) associated with lifestyle factors (e.g., smoking, alcohol consumption, diet, sleep deprivation, etc.) is increasing [5,6], which imposes a substantial economic burden on the Japanese economy. The status of national medical expenses published by the Japanese Ministry of Health, Labor and Welfare in 2019 indicates that national medical expenses are continuing to increase at a rate of 1 trillion yen per year [7].

One of the primary manifestations of the development of ICT in health is an increase in the number of available mHealth applications [8]. mHealth has been defined as “the use of mobile and wireless technologies to support the achievement of health objectives” [9]. During the COVID-19 pandemic, many countries worldwide faced a severe shortage of healthcare resources [10]. At the same time, most countries implemented prevention policies such as lockdowns and self-quarantining. In such a situation, researchers have been compelled to identify effective online treatment modalities. Therefore, in the last few years, mHealth has become an important complementary tool to cope with the shortage of medical resources [11,12].

In past studies, mHealth apps have been shown to positively impact individuals’ health [13,14]. However, in practice, the usage of health management apps is not high [15]. Even in Japan, a country with a high smartphone penetration rate, the usage rate of health management/exercise data services among Internet services for daily use is only 21.6% [1]. Many current studies focused on identifying the factors influencing mHealth adoption behaviors tend to focus on older adults [16,17] or patients who have an immediate need for personal health management [18,19]. However, the primary goal of mHealth is not only to provide health management after a health problem has been identified but also disease prevention through active self-management.

Young adults in Japan are critical potential users of mHealth because of two main reasons. First, the Japan White Paper on Intelligence and Communication (2021) reported that 96.9% of young adults aged 18–29 (*n* = 228) reported the frequent use of smartphones or tablets [1]. Thus, young people are more likely to receive mHealth-related information. Second, smartphone overuse poses additional health risks [20]. In the current era of frequent smartphone use, it is difficult to limit the use of smartphones, and therefore, preventing health risks through self-health management via smartphones seems to be an effective strategy. However, there is currently insufficient theoretical research supporting mHealth adoption among young adults in Japan. Therefore, this study aimed to examine the factors that influence the adoption of mHealth among young people in Japan.

## 2. Literature Review

It has been well-established that traditional technology acceptance models are equally applicable to the mHealth domain, such as the technology acceptance model (TAM) [21], unified theory of acceptance and use of technology (UTAUT) [22], and UTAUT2 [23], among others. For example, early research found that performance expectancy, effort expectancy, and facilitating conditions are direct factors in the adoption of medical information technology [24]. Hoque demonstrates that perceived usefulness, perceived ease of use, subjective norms, and trust are critical to mHealth adoption [25]. Binyamin and Zafar systematically reviewed 49 papers on mHealth app TAMs and found that perceived usefulness and perceived ease of use are the key factors influencing behavioral intention [26]. It is important to emphasize here that since effort expectancy and performance expectancy in the UTAUT model are considered to represent the same constructs as perceived ease of use and perceived usefulness, many papers combined these similar constructs into one term [26,27]. Yu et al. systematically analyzed 43 pieces of literature on e-health acceptance among older adults and similarly confirmed the importance of perceived usefulness and perceived ease of use [27]. Despite the maturity of these theoretical frameworks, the findings are inconsistent regarding the interpretation of mHealth adoption. For example, Macedo’s study of 278 Portuguese elderly people found that effort expectancy was an important factor influencing behavior intention [28], while Alam et al. found that effort expectancy did not influence behavior intention [29]. The results of Quaosar et al.’s questionnaire survey of 245 Bengali individuals indicated no significant effect of facilitation conditions on the intention to use health services [30], whereas, in Nisha et al.’s study comprising 927 Bengali individuals, facilitation conditions were found to be the most direct factor influencing mHealth adoption behavior [31].

Duarte and Pinho‘s study concluded that mHealth adoption is indeed a complex phenomenon influenced by the interaction of product characteristics, user characteristics, and psychological dimensions [32]. Therefore, researchers have been endeavoring to combine theoretical models or constructs for theory extension. For example, Luo et al. explored the psychological mechanisms of users’ continued use of mHealth apps from the perspective of the protective motivation theory and network externalities and confirmed that network externalities significantly impact users’ attitudes [33]. The protection motivation theory examines the factors influencing changes in health behavior from the perspective of safety motivation [34,35]. Network externalities, however, represent the effects or values that users obtain from other users using the same or similar products. Guo et al. extended the elaboration likelihood model (ELM), a fine-grained mimetic model widely used in e-commerce, by adding health consciousness as a moderating variable and applying it to the mHealth domain [36].

## 3. Research Model and Hypotheses

The theoretical foundation of this study is based on the well-established technology acceptance model, UTAUT, which was developed by Venkatesh et al. and was found to outperform the explanatory power of eight existing models, including the TAM, theory of rational action, motivation model, theory of planned behavior (TPB), TAM and TPB combined, personal computer use model, diffusion of innovation (DOI) theory, and social cognitive theory (SCT) [37]. This study is unique in that it extends the traditional UTAUT model framework by introducing three constructs, perceived risk, trust, and health consciousness, to the original model structure by combining the findings of previous studies. The revised UTAUT model used in this study included eight factors to evaluate behavior intention: health consciousness (HC), trust (TR), social influence (SI), perceived risk (PR), performance expectancy (PE), facilitating conditions (FC), effort expectancy (EE), and behavior Intention (BI).

Based on Duarte’s findings [32], this study examined Japanese adolescents’ behavioral intentions toward mHealth in terms of four aspects: personal characteristics, environmental characteristics, usage conditions, and subjective perceptions. Figure 1 illustrates our research model.

This study uses structural equation modeling to measure the factors that influence Japanese youths’ behavioral use of mHealth. Specifically, we used the psychological dimensions of subjective perceptions (PR, TR, PE, and EE) as the direct pathways of influence on behavioral intention. Personal characteristics (HC), environmental characteristics (SI), and conditions of use (FC) were used as external pathways indirectly influencing behavioral intention.

### 3.1. Perceived Risk

In this study, PR was defined as the extent to which people judge the possible adverse outcomes of mHealth. Some studies showed that an increase in PR has a direct negative impact on behavioral intentions regarding mHealth [38]. Foremost among these is the risk of privacy disclosure that health services related to information technology may pose to users [39]. Therefore, we propose the following hypothesis:

**Hypothesis** **1** **(H1).** 
*Perceived risk negatively affects behavioral intentions regarding mHealth.*


### 3.2. Trust

Trust is an important factor in attracting and retaining users [40]. Studies have shown that users are likely to adopt mHealth when they believe that it can deliver value [41]. Furthermore, given the perceived risks associated with mHealth, trust in policymakers, technology developers, and scientific researchers is equally important [42]. Therefore, trust has been defined as the degree of users’ trust in mHealth-related scientific research, governmental decisions, and technology developers. We proposed the following hypothesis:

**Hypothesis** **2** **(H2).** 
*Trust positively influences behavioral intentions regarding mHealth.*


### 3.3. Performance Expectancy

Performance expectancy is defined as the degree to which a person perceives that a new technology or service is useful for improving job performance [43]. In this study, PE indicates the subjective perceptions of the usefulness of mHealth for health management. Numerous studies have confirmed that PE is the most important factor influencing patients’ use of health management applications [44,45,46]. We, therefore, proposed the following hypothesis:

**Hypothesis** **3** **(H3).** 
*Performance expectancy has a positive effect on behavioral intentions regarding mHealth.*


### 3.4. Effort Expectancy

Effort expectancy is defined as the subjective perception of the difficulty of a technique [44]. In this study, EE indicates how easy or difficult it is for people to use mHealth applications or acquire knowledge of mHealth use. Studies demonstrated that people’s willingness to adopt mHealth technology increases when it is easy to use [47,48,49]. Especially regarding older adults, EE is often the most critical influencing factor [50]. Therefore, we proposed the following hypothesis:

**Hypothesis** **4** **(H4).** 
*Effort expectancy has a positive effect on behavioral intentions regarding mHealth.*


### 3.5. Health Consciousness

In this study, health consciousness was defined as the degree to which individuals value their health. In past studies, HC has been shown to have a direct influence on health attitudes and behaviors [51]. The results of Meng’s study indicated that HC reinforced the positive relationship between mobile health credibility and the intention to use mHealth. This finding also confirmed the validity of HC as a personal attribute in the peripheral pathway [52]. Furthermore, HC as a peripheral pathway has been shown to positively affect perceived usefulness [53,54]. Furthermore, health-conscious individuals tend to acquire health-related knowledge more frequently and actively [55]. Therefore, we hypothesized that health-aware individuals have higher risk-averse abilities because they are more aware of selecting the optimal mHealth product or service for them and know how to protect their privacy while using mHealth. Therefore, we hypothesize that:

**Hypothesis** **5a** **(H5a).** 
*Health consciousness has a negative effect on the perceived risk of mHealth.*


**Hypothesis** **5b** **(H5b).** 
*Health consciousness has a positive effect on trust related to mHealth.*


**Hypothesis** **5c** **(H5c).** 
*Health consciousness has a positive effect on the perceived usefulness of mHealth.*


### 3.6. Social Influences

In this study, we considered SI as an environmental characteristic of mHealth. SI was defined as the influence of the perceptions of others on the willingness to use mHealth. In the traditional UTAUT model, SI directly influences behavioral intention. However, complex psychological influences are often at play in influencing the relationship between SI and behavioral intention in terms of subjective consciousness. For example, social influence indirectly influences behavioral intention through the mediating role of perceived usefulness [43,56]. Moreover, in the field of psychology and neuroscience, it was found that most human choices influence trust preferences [57] and risk perceptions. When individuals face uncertainty, they incur PR for unknown consequences [58], and SI can reduce the PR caused by uncertainty [59]. Therefore, we hypothesize that:

**Hypothesis** **6a** **(H6a).** 
*Social influence has a negative effect on the perceived risk of using mHealth.*


**Hypothesis** **6b** **(H6b).** 
*Social influence has a positive effect on trust related to mHealth.*


**Hypothesis** **6c** **(H6c).** 
*Social influence has a positive effect on the perceived usefulness of mHealth.*


### 3.7. Facilitation Conditions

The facilitation condition is defined as the extent to which individuals perceive that the existing mHealth infrastructure can support mHealth use [37]. In this study, we used the question items from the UTAUT model [33] and applied them to mHealth. Specifically, facilitation conditions indicate the resources necessary to use mHealth, knowledge regarding mHealth, and interchangeability with other technologies. Some studies have indicated that when better facilitation conditions are available, users perceive that using the technology does not require much effort, which enhances behavioral intention [60,61]. Therefore, we hypothesize that:

**Hypothesis** **7** **(H7).** 
*Facilitation condition has a positive effect on effort expectancy regarding mHealth.*


## 4. Methods

### 4.1. Questionnaire Design

All question items in the survey were adopted or adapted from scales validated in prior studies. We adopted the constructs (PE, EE, SI, FC, BI) used in UTAUT proposed by Venkatesh et al. [37]. The 5 items measured by HC were from Guo et al. [36]. The PR-related question items were from Deng et al. [62]. TR was measured using the method of Farid et al. [63]. Because the research contexts of the referenced questionnaires vary, the question item was modified to suit our research context. Since the referenced scale was in English and the respondents were Japanese, the questionnaire was translated by two native Japanese speakers. In order to improve the validity of the questionnaire, a pretest was administered to 27 Japanese university students to ensure that the respondents could clearly understand the question items. A 7-point Likert scale was used for all items, with “1” representing “strongly disagree” and “7” representing “strongly agree.” Details of the questionnaire are in the Multimedia Appendix A.

A brief introduction to mHealth was provided to the participants before the survey began, covering the concept of mHealth, its application, advantages, and current shortcomings. A short introduction to mHealth is necessary for two reasons: (1) for people who have no experience with mHealth, this introduction helps them understand what mHealth is; (2) there may be cases where someone has used the experience of mHealth but is unaware of it due to a lack of understanding of the concept. The presentation was translated into Japanese by two native Japanese speakers. A native Japanese speaker delivered a 5-min presentation to the participants. We added a QR code on the last page of the presentation material to enable the participants to access the questionnaire. Participants could scan the QR code and fill out the questionnaire.

### 4.2. Data Collection

We created a questionnaire through Google Forms and surveyed university students in a classroom at a Japanese university. The necessary information pertaining to informed consent was included on the first page of the Google form. Respondents were required to provide their consent before they could answer the questionnaire. The survey did not offer any financial incentives, and respondents willingly volunteered to take the survey. Due to the impact of COVID-19, the university’s courses were conducted both online and offline. Considering that some students may be unable to participate directly in class, data collection was conducted in two ways: Method (1) Students responded to the questionnaire directly through the QR code presented in the classroom session; Method (2) Since the university uses the Manaba cloud-based collaborative learning system, students could access the classroom page and watch the recorded and uploaded video and scan the QR code that appears at the end of the video to answer the questionnaire. In order to collect as much data as possible, this study implemented two data collection sessions. The first was implemented from 10 December 2021 to 20 December 2021, and 184 responses were collected; the second was implemented from 20 April 2022 to 27 April 2022, and 104 responses were collected, amounting to a total of 288 responses. At the end of the questionnaire, we included question items asking whether it was the first time the respondent was answering the questionnaire to filter out the possibility of duplicate responses. Furthermore, secondary filtering of duplicate responses was performed by filtering the access emails of Google Forms. After filtering out duplicate responses, we obtained 233 valid responses with a response rate of 80.9%.

### 4.3. Data Analysis

Descriptive statistical analysis of the data used in this study was performed using the IBM SPSS Statistics (v. 27) software. The reliability and validity of the questionnaire structure and multicollinearity were tested using Smartpls (v. 3.3.5) software. Due to the small sample size, the partial least squares structural equation modeling (PLS-SEM) of Smartpls (v. 3.3.5) was used to validate the proposed model. PLS-SEM is widely used in the field of social sciences as a data analysis method for assessing complex causal models between potential variables [64]. An advantage of the PLS-SEM is that it supports small samples of the model prediction. The “10-times rule” was used as an estimation method for the minimum sample size, i.e., the sample size is greater than 10 times the maximum number of internal or external model links pointing to any potential variable in the model [65]. The total sample size of 233 valid responses obtained in this study meets the requirement for model validation.

## 5. Results

Table 1 shows the results of the descriptive statistics. The sample characteristics of this study were relatively evenly distributed between male and female participants. All the participants were young adults and current college students who were well-experienced in smartphone use. Participants who had mHealth experience accounted for only 32.19% of the total sample.

Before measuring the results, we first conducted a systematic evaluation of the questionnaire. First, we tested the factor loadings of all question items to ensure the convergent validity of the constructs. Furthermore, the variance inflation factor (VIF) of each item was tested to avoid multicollinearity between items. The results indicated that the factor loadings of FC1 were below the recommended threshold of 0.7, while the VIF of PR2 was higher than 5. Therefore, we removed these two problematic items in the subsequent analysis. Table 2 shows the comparison of the detection results before and after removing FC1 and PR2. After removing the non-conforming items, all items indicated a VIF of less than 5, factor loadings above 0.7, and *p* < 0.001.

In order to obtain robust results, we further measured the convergent validity of the model by evaluating the average variance extracted (AVE). Cronbach’s alpha coefficient and composite reliability (CR) were calculated to ensure construct reliability, and whether the square root of AVE for any construct was higher than the correlation with other constructs was examined to demonstrate discriminant validity. The results are shown in Table 3 and Table 4. Cronbach’s alpha coefficient and composite reliability (CR) exceeded the recommended threshold of 0.7, thus confirming the reliability of the constructs. The correlation between the factors was much smaller than the square root of the mean variance extracted for each factor, which confirms the discriminant validity.

Finally, we analyzed the structural model and tested the hypotheses. The path coefficients, as well as their significance, were calculated using the bootstrap method using a random subsample from the original data. In order to ensure the stability of the results, we set 5000 subsamples. The results are shown in Figure 2 and Table 5. Specifically, TR had significant positive effects on BI, PE had significant positive effects on BI, and EE had significant positive effects on BI, thereby supporting H2, H3, and H4. HC and SI as external pathways had significant positive effects on TR and PE, thereby supporting H5b, H5c, H6b, and H6c. FC had significant positive effects on EE, thereby supporting H7. However, PR had no significant effect on BI and, therefore, does not support H1. Furthermore, neither HC nor SI as external pathways were found to have a statistically significant effect on PR, and therefore, H5a and H6a were not supported. The R^2^ of behavioral intention is 0.44, which indicates that the factors included in this model are sufficient to explain the change in behavioral intention.

## 6. Discussion

The current study aimed to examine the factors affecting the behavioral intention of young adults in Japan regarding the adoption of mHealth. Based on previous studies, we adapted and extended the traditional UTAUT model using a more complex model structure by including the psychological dimension of subjective perceptions as a direct influence on behavioral intentions and as a mediator of personal characteristics, environmental characteristics, and conditions of use. The main findings of the study are summarized as follows.

First, the results of the descriptive statistics suggest that mHealth adoption remains low even among young Japanese adults with high smartphone usage rates. Only 32.19% of the total sample had prior experience with mHealth use. This finding further supports the need to explore the factors that influence mHealth adoption among young Japanese adults.

Second, we identified TR, EE, and PE as direct influential factors in the subjective perceptions that affect mHealth use among Japanese young adults. Among them, performance expectation showed the highest path coefficient (β = 0.341, *p* < 0.001). This finding is consistent with existing studies [26,27]. Therefore, for product designers, the primary consideration should be how to improve the usefulness of mHealth services. Trust was the second direct path of influence on behavioral intentions (β = 0.289, *p* < 0.001). This finding suggests that when people perceive more trust in government decisions, researchers, and product developers, it significantly and positively influences their behavioral intention to use mHealth. This finding has some important practical implications. For product designers, this finding provides additional insight into the importance of increasing trust in the design and development of mHealth products in addition to focusing on the value created by the product itself in the promotion of mHealth. In addition, it is also important that government regulations be improved and that academic researchers be trusted. Therefore, it is important for policymakers to improve regulations so that users can subjectively perceive the government’s positive attitude toward mHealth development. At the same time, encouraging more researchers to participate in mHealth research by providing research funding support at the policy level can help increase user trust in mHealth services. Effort expectation had the least significant positive effect on behavioral intention (β = 0.177, *p* < 0.01). This finding can be explained by the fact that young Japanese adults have a high degree of experience in using information technology and, therefore, demonstrate a higher acceptance of the perceived ease of use of smartphone devices. Therefore, relative to PE and trust, EE had a smaller effect on behavioral intention.

Third, regarding the improvement of the subjective perceptions of these psychological dimensions, the peripheral path suggests some promising avenues. SI showed a significant positive impact on TR (β = 0.432, *p* < 0.001) and PE (β = 0.332, *p* < 0.001). This finding validates similar findings in the field of psychology [57]. People often tend to change their behavior and opinions to conform to social norms [66]. On the practical side, mHealth product managers can organize campaigns, such as choosing people who are socially influential to young Japanese people to promote the product and paying attention to the role of TR and PE. Social attributes can also be designed for the mHealth product so that more people can access mHealth. The significant positive effect of HC on TR (β = 0.279, *p* < 0.001) and PE (β = 0.261, *p* < 0.001) was an equally critical finding of this study. A greater degree of HC implies a willingness to actively pursue a healthy lifestyle [67]. Thus, relative to those with low HC, people with high HC are more willing to proactively obtain mHealth-related information and learn about the value of mHealth for their well-being. Therefore, in addition to the promotion of mHealth itself, mHealth managers should also pay attention to promoting the benefits of a healthy lifestyle to improve the HC of young Japanese. FC showed a significant positive effect on EE (β = 0.432, *p* < 0.001), which indicates that when people have more resources and knowledge to use mHealth, they perceive using mHealth as easy.

However, no significant effect of PR on behavioral intention was found in this study. Furthermore, the effects of HC and SI on PR were not significant. The review of past studies found inconsistent findings regarding the effect of PR on behavioral intentions. For example, Pan’s study of 744 Chinese nurses found no significant effect of PR on behavioral intentions regarding mHealth [22]. Whereas performance risk and privacy risk were found to have a significant negative impact on mHealth adoption in Deng et al.’s study, legal issues were not associated with mHealth adoption intentions [62]. Thus, there may be a more complex relationship between PR and behavioral intention pertaining to mHealth, which needs to be further explored in future studies.

### 6.1. Theoretical Implications

This study provides theoretical contributions to the popularization of mHealth technology from the following two aspects.

First, our study makes a significant theoretical contribution to the popularization of mHealth in Japan. A bibliometric analysis of mHealth from 2000 to 2020 showed that the number of Japanese publications was only 175 out of 12,593 mHealth-related publications in the web of science by 2020 [2]. This result indicates that research on mHealth in Japan is still very limited. Furthermore, to the best of our knowledge, there is a lack of research on the mHealth technology acceptance theory in Japan. Currently, mHealth is maturing in terms of technology and is in the stage of social application [2]. However, the diffusion of technology requires theoretical support. This study explains the mechanisms that influence the adoption of mHealth among young people in Japan based on an extended UTAUT model. It fills a gap in the lack of theory to popularize mHealth in Japan.

Second, we made equally critical theoretical contributions in adapting and extending the original UTAUT model to fit mHealth. Although previous research has extended the UTAUT model, the extended constructs have often been used as direct influence factors on BI, ignoring the complexity of influencing people’s behavioral intentions. The present study categorized the demonstrated constructs from previous studies. The UTAUT model was reintegrated with the subjective perception of the psychological dimension as the direct influence path of BI and other factors as external paths influencing the psychological dimension to enrich the theoretical framework.

### 6.2. Practical Implications

Our study likewise provided some practical implications.

First, this study provides support for mHealth entrepreneurs in Japan. The theoretical model of this study can help product designers clarify the direction of technology development and contribute to a more comprehensive provision of mHealth services.

Second, this study also provided practical implications for policymakers. National policies are critical to the development of technology. China’s emphasis on medical Informatization has driven the popularity of mHealth in China’s healthcare field [68]. This study can help policymakers to grasp the psychological characteristics of young people in Japan and help them to decide the direction of policymaking.

### 6.3. Limitations

Although the present study revealed important findings, some limitations need to be taken into account. First, the current study sample was collected at only one Japanese university and had a small sample size. This limitation may lead to biases since the participants were all young Japanese adults who were students by occupation. Therefore, in future studies, the sample size should be expanded to include different occupational and educational backgrounds as well as diverse age groups. Second, there may be differences in the influence of subjective perceptions on behavioral intentions between those who have prior experience with mHealth use and those who have no experience with mHealth use. We were unable to validate this distinction, given the large difference in sample size between the two cohorts in this study. Therefore, in future studies, the sample size needs to be expanded to group the samples by whether or not they have experience with mHealth use to explore the differences in the effects of mHealth experience on behavioral intention.

## 7. Conclusions

This study examined the factors influencing the behavioral intentions of young Japanese adults to adopt mHealth in terms of four dimensions: personal characteristics, environmental characteristics, conditions of use, and subjective perceptions. Our findings indicate that PE, TR, and EE are direct influencers of behavioral intention toward mHealth adoption. Furthermore, behavioral intention can be influenced through increased HC, SI, and FC. The findings of this study have important theoretical implications for policymakers as well as product designers and developers for the further diffusion of mHealth adoption in the future.

## Figures and Tables

**Figure 1 ijerph-19-15156-f001:**
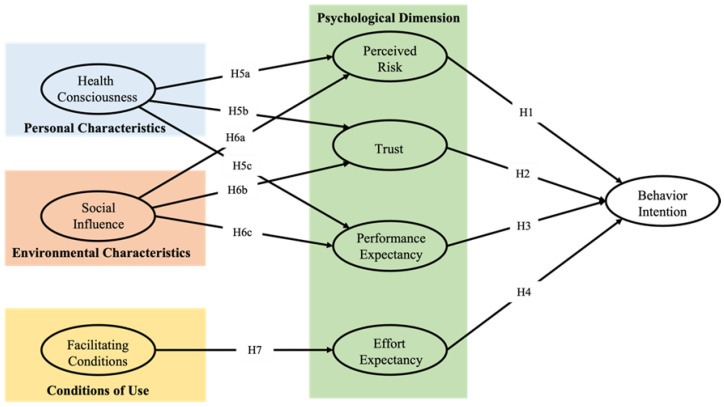
Revised model to be tested for hypotheses (H).

**Figure 2 ijerph-19-15156-f002:**
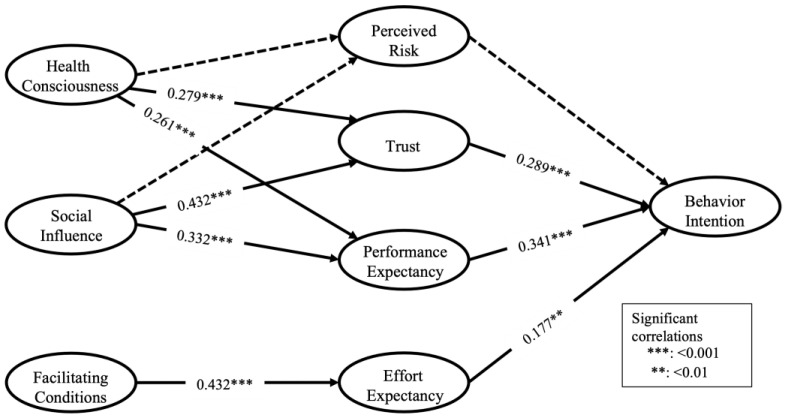
Standardized estimates for the revised model.

**Table 1 ijerph-19-15156-t001:** Descriptive statistics of participants (*n* = 233).

Item	Category	*n*	%
Sex	Male	136	58.37%
	Female	97	41.63%
Age	≤20	183	78.54%
	21–30	50	21.46%
Job	Student	233	100%
Smartphone Use (Year)	1–3	12	4.76%
	4–7	153	60.71%
	8–10	77	30.56%
	≤10	10	3.97%
mHealth Use	Yes	75	32.19%
	No	158	67.81%

**Table 2 ijerph-19-15156-t002:** Outer loading and VIF test of questionnaire items.

Items	Before			After				
	Path Coefficients	T Statistics	*p*-Values	VIF	Path Coefficients	T Statistics	*p*-Values	VIF
BI1	0.908	54.233	0.000	2.755	0.908	53.321	0.000	2.755
BI2	0.929	90.595	0.000	3.223	0.929	89.671	0.000	3.223
BI3	0.917	61.460	0.000	2.978	0.917	61.429	0.000	2.978
EE1	0.791	21.048	0.000	1.792	0.786	19.867	0.000	1.792
EE2	0.841	42.214	0.000	1.986	0.843	42.752	0.000	1.986
EE3	0.866	42.193	0.000	2.177	0.867	41.652	0.000	2.177
EE4	0.825	28.987	0.000	1.866	0.825	28.976	0.000	1.866
FC1	0.398	3.662	0.000	1.045	-	-	-	-
FC2	0.862	30.356	0.000	1.528	0.874	31.762	0.000	1.463
FC3	0.836	19.084	0.000	1.475	0.894	35.723	0.000	1.463
HC1	0.807	25.914	0.000	2.007	0.808	26.257	0.000	2.007
HC2	0.819	23.996	0.000	2.361	0.819	24.243	0.000	2.361
HC3	0.727	15.050	0.000	1.522	0.726	15.041	0.000	1.522
HC4	0.791	17.658	0.000	1.920	0.791	17.752	0.000	1.920
HC5	0.829	24.147	0.000	2.345	0.828	25.105	0.000	2.345
PR1	0.905	15.750	0.000	3.326	0.911	13.542	0.000	2.761
PR2	0.943	20.172	0.000	5.688	-	-	-	-
PR3	0.929	20.135	0.000	4.781	0.930	14.923	0.000	4.017
PR4	0.891	15.296	0.000	4.183	0.892	10.818	0.000	3.893
PR5	0.905	17.385	0.000	4.388	0.915	12.737	0.000	4.350
PE1	0.789	25.011	0.000	2.039	0.789	25.470	0.000	2.039
PE2	0.817	27.092	0.000	2.322	0.817	26.067	0.000	2.322
PE3	0.871	44.909	0.000	2.653	0.871	44.703	0.000	2.653
PE4	0.863	45.664	0.000	2.653	0.863	46.066	0.000	2.653
PE5	0.862	40.700	0.000	2.615	0.862	42.107	0.000	2.615
SI1	0.883	47.112	0.000	2.351	0.883	46.616	0.000	2.351
SI2	0.926	69.110	0.000	3.285	0.926	68.231	0.000	3.285
SI3	0.911	50.308	0.000	2.700	0.911	50.667	0.000	2.700
TR1	0.740	16.082	0.000	1.638	0.740	16.378	0.000	1.638
TR2	0.786	25.164	0.000	1.822	0.786	25.083	0.000	1.822
TR3	0.814	23.183	0.000	2.364	0.814	23.333	0.000	2.364
TR4	0.851	32.468	0.000	2.944	0.851	32.218	0.000	2.944
TR5	0.816	30.719	0.000	2.279	0.816	31.109	0.000	2.279

**Table 3 ijerph-19-15156-t003:** Correlation matrix after removing non-conforming items.

Factors	Correlation							
	BI	EE	FC	HC	PR	PE	SI	TR
BI	0.918							
EE	0.457	0.831						
FC	0.355	0.432	0.884					
HC	0.364	0.357	0.245	0.795				
PR	0.585	0.486	0.418	0.385	0.841			
PE	0.109	0.013	0.122	0.105	0.071	0.912		
SI	0.389	0.319	0.436	0.373	0.429	0.138	0.907	
TR	0.547	0.392	0.424	0.440	0.541	0.235	0.536	0.802

**Table 4 ijerph-19-15156-t004:** Validity and reliability tests.

	Before				After			
Factors	Cronbach’s Alpha	rho_A	Composite Reliability	Average Variance Extracted (AVE)	Cronbach’s Alpha	rho_A	Composite Reliability	Average Variance Extracted (AVE)
BI	0.907	0.908	0.941	0.842	0.907	0.908	0.941	0.842
EE	0.851	0.859	0.899	0.691	0.851	0.861	0.899	0.690
FC	0.516	0.619	0.758	0.533	0.720	0.724	0.877	0.781
HC	0.854	0.854	0.896	0.633	0.854	0.854	0.896	0.633
PR	0.896	0.903	0.923	0.707	0.896	0.903	0.923	0.707
PE	0.952	0.981	0.963	0.837	0.935	0.987	0.952	0.832
SI	0.892	0.895	0.933	0.822	0.892	0.895	0.933	0.822
TR	0.861	0.861	0.900	0.644	0.861	0.861	0.900	0.644

**Table 5 ijerph-19-15156-t005:** Results of the hypothesis test.

	Path Coefficient	*p*-Values	Hypotheses
EE → BI	0.177	0.003	Supported
FC → EE	0.432	0.000	Supported
HC → PE	0.261	0.000	Supported
HC → PR	0.062	0.406	Unsupported
HC → TR	0.279	0.000	Supported
PE → BI	0.341	0.000	Supported
PR → BI	0.014	0.797	Unsupported
SI → PE	0.332	0.000	Supported
SI → PR	0.115	0.168	Unsupported
SI → TR	0.432	0.000	Supported

## Data Availability

Raw data were generated at Ritsumeikan Univerity. Derived data supporting the findings of this study are available from the corresponding author KK on request.

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
