# Peer review of "Social Acceptance of Mobile Health among Young Adults in Japan: An Extension of the UTAUT Model"

_ijerph, 2022, doi:10.3390/ijerph192215156_

Round 1

Reviewer 1 Report

The rapid development of digital devices and concern for health care has led to a growing interest in mHealth. Echoing this, this study analyzes the factors that may influence the adoption of this trend among Japanese youth. This is an original study and of considerable scientific relevance, since the identification of such elements may be decisive for its diffusion among this group. However, there are some aspects that need to be reviewed and modified:

-Authors are encouraged to indicate the number of participants in the study in the abstract.

-The literature review is somewhat scarce, so we suggest expanding this section.

-On the other hand, it is necessary to revise the manuscript and adapt the citation and reference styles to the journal’s standards (e.g., lines 103-107).

-In the Method section (lines 196-197) it is stated: “All question items in the survey were adopted or adapted from scales validated in prior studies”. It would be useful to indicate what these prior studies are. In addition, the authors are encouraged to explain why and how this adaptation was carried out.

-It would also be advisable to adapt the format of the tables to that proposed by the journal.

-The Discussion section is too brief and, in some cases, consists of a repetition of the findings shown in the previous section. The authors are actively encouraged to expand this section and to discuss the results more intensively.

-The Conclusions are very few and limited. For example, lines 359-361 state: “The findings of this study have important theoretical implications for policymakers as well as product designers and developers for the further diffusion of mHealth adoption in the future”. However, it would be interesting for the authors to indicate explicitly what are the theoretical and practical implications derived from the study. Therefore, it is suggested that the authors expand the Conclusions section and point out the implications derived from the research.

Author Response

Dear Editor and Reviewers,

On behalf of all the co-authors, I would like to sincerely thank you for giving us the opportunity to submit a revised draft of our manuscript to " International Journal of Environmental Research and Public Health." Further, we are deeply thankful for the time and efforts spent by the experienced reviewers to point out the opportunities for developing the manuscript further. We highly appreciate the reviewers' expertise and comments; it has indeed added a lot to the manuscript, and we feel grateful for such detailed feedback from the reviewers.

After carefully reviewing all the constructive feedback and comments by the reviewers, we have been able to incorporate many of changes suggested by the reviewers. We have also utilized the "track changes" function in "Microsoft Word," as you have kindly advised, making it easier for further revisions.

The manuscript has been rechecked and the necessary changes have been made in accordance with the reviewers’ suggestions. The responses to all comments have been prepared and attached herewith.

We hope that our responses meet your expectations and your intentions.

We highly appreciate your cooperation.

Warm Regards,

Kota Kodama, PhD

Graduate School of Technology Management, Ritsumeikan University

2-150, Iwakura-cho, Ibaraki, Osaka, 567-8570, Japan

+81-72-665-2448

[email protected]

ID

Comments and Suggestions

Response

Reviewer 1

1) Authors are encouraged to indicate the number of participants in the study in the abstract.

Thank you very much for this important comment. An introduction to the number of respondents has been added to the abstract.

Reviewer 1

2) The literature review is somewhat scarce, so we suggest expanding this section

Thanks for your suggestion. We appropriately added the introduction of some prior studies in the literature review section.

Reviewer 1

3) On the other hand, it is necessary to revise the manuscript and adapt the citation and reference styles to the journal’s standards (e.g., lines 103-107).

Thanks for your thoughtful comment. We reconfirmed the ijerph-template and referenced the citation format of other papers published in MDPI. Inappropriate places have been revised to suit journal standards.

Reviewer 1

4) In the Method section (lines 196-197) it is stated: “All question items in the survey were adopted or adapted from scales validated in prior studies”. It would be useful to indicate what these prior studies are. In addition, the authors are encouraged to explain why and how this adaptation was carried out.

Thanks for your constructive comments. We think your suggestions will improve readers' understanding of this study. Therefore, a description of previous studies related to the questionnaire has been added to the Methods section.

Reviewer 1

5) It would also be advisable to adapt the format of the tables to that proposed by the journal.

Thanks for your thoughtful comment. We reconfirmed the ijerph-template and referenced the citation format of other papers published in MDPI. Inappropriate places have been revised to suit journal standards.

Reviewer 1

6) The Discussion section is too brief and, in some cases, consists of a repetition of the findings shown in the previous section. The authors are actively encouraged to expand this section and to discuss the results more intensively.

Thank you very much for pointing out the shortcomings of the article. We full accept this valuable comment. We expand the Discussion section to provide a more in-depth understanding of the implications of this study.

Reviewer 1

7) The Conclusions are very few and limited. For example, lines 359-361 state: “The findings of this study have important theoretical implications for policymakers as well as product designers and developers for the further diffusion of mHealth adoption in the future”. However, it would be interesting for the authors to indicate explicitly what are the theoretical and practical implications derived from the study. Therefore, it is suggested that the authors expand the Conclusions section and point out the implications derived from the research.

We appreciate your time and effort in reviewing this section and giving detailed suggestions for improvement. Considering the brevity of the conclusion section, we describe the Theoretical Implications and Practical Implications of this study in detail in the Discussion section. Hope it can give you a better reading experience.

Reviewer 2 Report

The revised UTAUT model is used to study influence of Japanese youths' behavioral use mHealth applications. The suitable survey was prepared and distributed among students studying at Japanese universities. A brief introduction to mHealth solutions was provided to the all  participants before the survey began. The idea of mHealth, its possibilities and also  advantages and shortcomings were presented. It is easy to note that results of surveys strongly depends on that presentations. More interesting will be respondents coming from students having some real contacts with  the mHealth applications. 

Besides,  in the paper there is another  mistake. Page 3, line 10-13, 9 factors is given instead of 8 ones. Extra one is Perceived Usefulness (PU), which is omitted in the model shown in Fig.1. Moreover in Table 2 and Table 3  the parameter PU is analyzed.  In Fig. 2 the parameter PE (Performance Expectancy)  is shown again. The same in Table A1? In consequence the consideration and results raise serious doubts.

Author Response

Dear Editor and Reviewers,

On behalf of all the co-authors, I would like to sincerely thank you for giving us the opportunity to submit a revised draft of our manuscript to " International Journal of Environmental Research and Public Health." Further, we are deeply thankful for the time and efforts spent by the experienced reviewers to point out the opportunities for developing the manuscript further. We highly appreciate the reviewers' expertise and comments; it has indeed added a lot to the manuscript, and we feel grateful for such detailed feedback from the reviewers.

After carefully reviewing all the constructive feedback and comments by the reviewers, we have been able to incorporate many of changes suggested by the reviewers. We have also utilized the "track changes" function in "Microsoft Word," as you have kindly advised, making it easier for further revisions.

The manuscript has been rechecked and the necessary changes have been made in accordance with the reviewers’ suggestions. The responses to all comments have been prepared and attached herewith.

We hope that our responses meet your expectations and your intentions.

We highly appreciate your cooperation.

Warm Regards,

Kota Kodama, PhD

Graduate School of Technology Management, Ritsumeikan University

2-150, Iwakura-cho, Ibaraki, Osaka, 567-8570, Japan

+81-72-665-2448

[email protected]

ID

Comments and Suggestions

Response

Reviewer 2

1)It is easy to note that results of surveys strongly depends on that presentations. More interesting will be respondents coming from students having some real contacts with the mHealth applications.

Thank you for your detailed response to this study. As you mentioned, the presentation is very important to improve the reader's understanding of mHealth to adequately answer to the questions. Therefore, for readers understanding, we have added the rationale for setting up the presentation in the text.

We fully agree that student respondents who have actual contact with mHealth are important research subjects. Considering that fewer people in our sample had experience with mHealth use, the need for the minimum sample size required for the model was not met. Therefore, we plan to expand the sample size in future studies by comparing groups of people with and without experience with mHealth use to capture more precisely the psychological characteristics of users' adoption of mHealth.

Reviewer 2

2) Besides, in the paper there is another mistake. Page 3, line 10-13, 9 factors is given instead of 8 ones. Extra one is Perceived Usefulness (PU), which is omitted in the model shown in Fig.1. Moreover in Table 2 and Table 3 the parameter PU is analyzed.  In Fig. 2 the parameter PE (Performance Expectancy) is shown again. The same in Table A1? In consequence the consideration and results raise serious doubts.

We appreciate your time and effort in reviewing this manuscript and pointing out the shortcomings of the article in such detail. As mentioned in lines 76-78, "Perceived Usefulness" and "Performance Expectancy" have similar structures, so we find that they are often combined into one item statistic in related literature reviews. For example, references [24, 25] are incorporated as "Perceived Usefulness." Considering that the theory on which this article is based is UTAUT, we think it is more appropriate to use "Performance Expectancy."

This is our mistake, the erroneous abbreviation "PU" was used in the data analysis for the items analyzed. We apologize for the confusion caused by your understanding of this article. The corresponding places have been modified and deleted.

Thanks again for your constructive comments

Round 2

Reviewer 1 Report

The manuscript has been significantly modified and improved. Therefore, it is considered suitable for publication.